# One-shot Entropy Minimization for Language Model Reasoning

**Zitian Gao**[1]  **Yilong Chen**[1]  **Haoming Luo**[1]  **Joey Zhou**[1]  **Bryan Dai**[1]

## Abstract

In this work, we propose One-shot Entropy Minimization (EM), a simple and fully unsupervised post-training approach that significantly improves reasoning and generation performance using only a single unlabeled data and approximately ten gradient steps. **To avoid data contamination, we pretrain a 7-billion-parameter language model from scratch with strictly decontaminated data**. Despite its extreme simplicity, one-shot EM yields substantial performance gains and improves reasoning abilities across a broad range of domains, including mathematical reasoning, logical reasoning, and coding. We further show that entropy minimization induces a characteristic right-skewed logit shift, amplifying high-probability tokens while suppressing low-probability tails, in contrast to reinforcement learning. Our findings suggest that entropy minimization primarily acts as a distribution shaping mechanism rather than a conventional learning process, offering an efficient and practical algorithm for post-training large language models.

## 1. Introduction

The post-training phase of large language models (LLMs) has advanced rapidly (Xie et al., 2025; Hu et al., 2025; Wang et al., 2025; Cui et al., 2025; Zeng et al., 2025), with models like DeepSeek-R1 (DeepSeek-AI et al., 2025), Kimi-K1.5 (Team et al., 2025), and OpenAI o-series (OpenAI, 2024; 2025) demonstrating remarkable reasoning abilities.

However, reinforcement learning (RL) typically requires large amounts of high-quality labeled data and carefully engineered rule-based rewards to maximize advantage signals and avoid reward hacking. In contrast, Entropy Minimization (EM) is a fully unsupervised approach: our study shows that a single unlabeled example and only 10 gradient steps

are sufficient to achieve significant, broad improvements in reasoning performance, converging orders of magnitude faster than RL. EM is based on two simple assumptions: (1) sampling in large language model generation is inherently stochastic, and (2) correct answers generally exhibit lower entropy than incorrect ones. We further show that EM and RL share the same objective—unlocking latent capabilities of pretrained models without introducing new knowledge (Liu et al., 2025)—and both operate through a process we term token reranking.

Recent studies suggest that the effectiveness of some unsupervised or lightly supervised RL methods may be affected by data contamination in base models (Wu et al., 2025). To ensure reliable conclusions, we pretrain a 7-billion-parameter model, Deconta-7B, entirely from scratch using a strictly decontaminated pipeline across pretraining, supervised fine-tuning, and RL. Under this controlled setting, Deconta-7B achieves consistent and significant improvements across mathematical reasoning, logical reasoning, and code generation tasks after One-shot EM, providing strong evidence for the effectiveness of our method.

Our main contributions are as follows:

- We propose *One-shot Entropy Minimization*, a surprisingly powerful and fully unsupervised method using just a single unlabeled data.

- We conduct an in-depth analysis of the effectiveness of One-shot EM, making it a highly reasonable approach. We find that it shares many core properties with RL, yet drives model behavior in the opposite direction when viewed through the lens of *logits shift*.

- We reveal that EM is a distribution shaping tool rather than a learning method by analyzing logit shift in Section 3.4 effect and the inconsistency in loss-reasoning curve over confidence in Section 3.6 of EM.

- To rigorously eliminate data contamination, we pretrain a large language model from scratch, **Deconta-7B**, on a strictly decontaminated corpus; experimental results on Deconta-7B fully demonstrate the effectiveness of One-shot EM.

---

[1]IQuest Research. Correspondence to: Bryan Dai <cb-dai@ubiquant.com>.

*Proceedings of the $43^{rd}$ International Conference on Machine Learning*, Seoul, South Korea. PMLR 306, 2026. Copyright 2026 by the author(s).

## 2. Method

### 2.1. Entropy Minimization Algorithm

Let $\mathcal{V}$ denote the vocabulary of a pretrained autoregressive language model $p_\theta$, parameterized by $\theta$. Given an input prompt $x$ (e.g., a question or problem description), the model autoregressively generates a response sequence $y = (y_1, y_2, \ldots, y_T)$ according to its current policy:

$$p_\theta(y \mid x) = \prod_{t=1}^{T} p_\theta(y_t \mid y_{<t}, x),$$

where $T$ is the length of the generated sequence.

Our core idea is to reduce the model's uncertainty over its own predictions by minimizing the token-level entropy at each generation step. The conditional entropy at time step $t$ is defined as:

$$H_t = -\sum_{v \in \mathcal{V}} p_\theta(v \mid y_{<t}, x) \log p_\theta(v \mid y_{<t}, x).$$

To avoid penalizing the prompt portion, we compute entropy only over the generated tokens. Let $T_{\text{prompt}}$ denote the number of tokens in the prompt $x$. Then the set of target positions is:

$$\mathcal{I} = \{t \mid t > T_{\text{prompt}}, \ y_t \neq \texttt{[PAD]}\}.$$

The overall EM loss for a single input $x$ is given by:

$$\mathcal{L}_{\text{EM}}(x; \theta) = \frac{1}{|\mathcal{I}|} \sum_{t \in \mathcal{I}} H_t.$$

This loss encourages the model to become more confident in its own predictions without relying on external supervision. Entropy Minimization loss is fully differentiable with respect to model parameters, with gradients resembling the score-function estimator found in entropy-regularized reinforcement learning. What's more, Entropy Minimization offers a closed-form objective, eliminating the need for external reward estimation or value baselines, thereby simplifying optimization while retaining the effectiveness of entropy-driven exploration and exploitation.

### 2.2. Gradient Estimation and Optimization Properties

We now analyze the gradient of the entropy minimization (EM) objective and clarify its relationship to policy gradient methods and entropy-regularized optimization.

Recall that at each generation step $t$, the conditional entropy

is defined as

$$H_t = -\sum_{v \in \mathcal{V}} p_\theta(v \mid y_{<t}, x) \log p_\theta(v \mid y_{<t}, x).$$

Since $p_\theta(\cdot \mid y_{<t}, x)$ is a differentiable function of $\theta$, the EM loss

$$\mathcal{L}_{\text{EM}}(x; \theta) = \frac{1}{|\mathcal{I}|} \sum_{t \in \mathcal{I}} H_t$$

admits an exact gradient without resorting to Monte Carlo reward estimation or surrogate objectives.

**Exact gradient form.** Taking the gradient of $H_t$ with respect to $\theta$ yields

$$\nabla_\theta H_t = -\sum_{v \in \mathcal{V}} \nabla_\theta p_\theta(v \mid y_{<t}, x) \left( \log p_\theta(v \mid y_{<t}, x) + 1 \right).$$

Using the identity $\nabla_\theta p_\theta(v) = p_\theta(v) \nabla_\theta \log p_\theta(v)$, this can be rewritten as

$$\nabla_\theta H_t = -\mathbb{E}_{v \sim p_\theta(\cdot \mid y_{<t}, x)} \\ \left[ \left( \log p_\theta(v \mid y_{<t}, x) + 1 \right) \nabla_\theta \log p_\theta(v \mid y_{<t}, x) \right].$$

This expression closely resembles a score-function (policy gradient) estimator, with

$$r(v) = -\left( \log p_\theta(v \mid y_{<t}, x) + 1 \right)$$

playing the role of a *self-induced reward*. Crucially, this reward is available in closed form and depends only on the model's own predictive distribution, rather than on any external supervision or task-specific signal.

**Connection to entropy-regularized RL.** In entropy-regularized reinforcement learning, one typically optimizes objectives of the form

$$\mathbb{E}[R] + \alpha H(\pi),$$

where entropy acts as a regularizer to encourage exploration. By contrast, EM directly optimizes the entropy term itself, but in the opposite direction. Rather than encouraging stochasticity, EM explicitly drives the policy toward lower-entropy, more concentrated predictive distributions.

### 2.3. Data Selection

Entropy minimization (EM) relies on the premise that the model's predictive uncertainty can serve as a meaningful training signal. However, not all input prompts are equally informative in this regard. As noted in prior work (Razin et al., 2025; Wang et al., 2025), certain prompts elicit deterministic behavior from the model (e.g., always correct or always incorrect), yielding limited gradient information under entropy-based objectives.

To address this, we adopt a variance-based data selection strategy. Specifically, we measure the variance of the model's *pass@k* accuracy across multiple samples and select prompts for which the model exhibits the highest behavioral variance. This targets inputs on the cusp of the model's capability—neither trivial nor impossible—making them ideal for entropy-driven optimization.

Given a prompt $x$, we draw $k$ independent samples from the model:

$$\mathcal{Y}^{(x)} = \left\{ y^{(1)}, y^{(2)}, \ldots, y^{(k)} \right\}, \quad y^{(i)} \sim p_\theta(\cdot \mid x).$$

We then compute the *pass@k* score as:

$$\text{pass@k}(x) = \frac{1}{k} \sum_{i=1}^{k} \mathbb{I}\left[ y^{(i)} \text{ is correct} \right],$$

where $\mathbb{I}[\cdot]$ is the indicator function for whether a sample is considered correct (via execution or string match).

We further compute the sample variance of this binary success variable:

$$\text{Var}_{\text{pass@k}}(x) = \frac{1}{k} \sum_{i=1}^{k} \left( \mathbb{I}\left[ y^{(i)} \text{ is correct} \right] - \text{pass@k}(x) \right)^2.$$

This variance quantifies the inconsistency of the model's predictions for a given input. A low variance indicates either high confidence in correctness (near-perfect success) or high confidence in failure (uniformly wrong), both of which are suboptimal for entropy minimization, as they lead to low-entropy posteriors that cannot be further improved.

We therefore define our data selection objective as:

$$x^* = \arg\max_{x \in \mathcal{D}} \text{Var}_{\text{pass@k}}(x),$$

where $\mathcal{D}$ denotes the unlabeled data pool.

This approach effectively prioritizes those prompts where the model exhibits the most behavioral uncertainty, making them "entropy-sensitive." Such prompts are empirically found to produce the largest entropy gradients and hence drive meaningful parameter updates under EM.

Intuitively, data with high *pass@k* variance suggests that the model's response distribution is straddling the decision boundary—sometimes correct, sometimes not—indicating a broad or multimodal predictive distribution. These are precisely the regions where entropy minimization is most impactful: it encourages the model to concentrate its probability mass on a consistent and (ideally) correct reasoning trajectory.

By contrast, if a model consistently answers a question correctly or incorrectly regardless of sampling, the entropy is either already minimal or optimization is ineffective. Thus, high-variance prompts provide the richest signal for improving model calibration and reasoning fidelity.

A sample from the NuminaMath (LI et al., 2024) dataset that meets the data filtering criteria is as follows:

---

**An example of a selected data**

**Problem**: The pressure $P$ exerted by wind on a sail varies jointly as the area $A$ of the sail and the cube of the wind's velocity $V$. When the velocity is $8$ miles per hour, the pressure on a sail of $2$ square feet is $4$ pounds. Find the wind velocity when the pressure on $4$ square feet of sail is $32$ pounds.

**Solution**: 12.8

---

The ablation results of the variance-based data selection strategy are shown in Table 1.

**Theoretical justification of variance-based data selection.** We formalize why prompts with high $\text{Var}_{\text{pass@k}}$ provide stronger training signals for entropy minimization.

**Setup.** Fix a prompt $x$ and sample $Y \sim p_\theta(\cdot \mid x)$. Let $C \in \{0, 1\}$ indicate correctness and $p(x) = \text{Pr}_\theta(C = 1 \mid x)$, the population analogue of pass@$k(x)$. Then $\text{Var}(C \mid x) = p(x)(1 - p(x))$.

**Lemma 1 (Correctness-induced entropy lower bound).** The sequence-level entropy satisfies

$$H(Y \mid x) \geq H(C \mid x) = h(p(x)),$$

where $h(\cdot)$ is the binary entropy.

*Proof.* By the chain rule, $H(Y, C \mid x) = H(C \mid x) + H(Y \mid C, x) = H(Y \mid x) + H(C \mid Y, x)$. Since $H(C \mid Y, x) \geq 0$, the claim follows.

**Proposition 1 (Optimality of high-variance prompts for EM).** The lower bound $H(C \mid x)$ is maximized when $p(x) = 1/2$, i.e., when $\text{Var}(C \mid x)$ is maximal. Moreover, higher $\text{Var}(C \mid x)$ implies larger achievable token-level entropy and stronger entropy-minimization gradients.

*Proof sketch.* $H(C \mid x) = h(p(x))$ is monotone in $p(x)(1 - p(x))$ on $[0, 1/2]$. By the entropy chain rule, $H(Y \mid x) = \sum_t H(Y_t \mid Y_{<t}, x)$, so higher sequence entropy implies higher average token entropy. For $p =$

*Table 1.* Comparison of different methods on math reasoning benchmarks (MATH500 (Lightman et al., 2023), MinervaMath (Lewkowycz et al., 2022), OlympiadBench (He et al., 2024), AMC23), logic reasoning benchmark (KK (Xie et al., 2024)), and code benchmark (MBPP (Austin et al., 2021)). To reduce randomness, each benchmark is evaluated under avg@8.

| Model | Dataset Size | Training Step | MATH500 | Minerva | OlympiadBench | AMC23 | KK | MBPP | Average Score |
|---|---|---|---|---|---|---|---|---|---|
| **OpenReasoner-Zero-7B** (Hu et al., 2025) | 129k | 600+ | 79.2 | 31.6 | 44.0 | 47.0 | 27.2 | 78.3 | 51.1 |
| **SimpleRL-Zoo** (Zeng et al., 2025) | 24K | 4000 | 76.8 | 30.9 | 39.4 | 55.3 | 17.4 | 75.9 | 49.3 |
| **Prime-Zero-7B** (Cui et al., 2025) | 230K | 240 | 83.8 | 36.0 | 40.9 | 62.7 | 4.4 | 41.0 | 44.8 |
| **Oat-Zero-7B** (Liu et al., 2025) | 12K | 300 | 80.0 | 30.1 | 41.0 | 62.7 | 2.2 | 58.7 | 45.8 |
| **RLVR** (Wang et al., 2025) | 1.2 K | 1000 | 78.6 | 33.8 | 41.6 | 62.5 | 12.4 | 54.5 | 47.2 |
| **RLVR 16-shot** | 16 | 1000 | 77.8 | 35.3 | 39.9 | 62.2 | 2.4 | 58.5 | 46.0 |
| **RLVR 1-shot** | 1 | 1000 | 78.6 | 36.0 | 43.7 | 61.9 | 5.8 | 60.3 | 47.7 |
| **Fully Decontaminated Models** | | | | | | | | | |
| **Deconta-7B-SFT** | NA | NA | 88.6 | 24.0 | 55.0 | 86.1 | 0.8 | 58.2 | 52.1 |
| **+ 1-shot EM** | 1 | 10 | $91.2^{\uparrow2.6}$ | $30.3^{\uparrow6.3}$ | $58.8^{\uparrow3.8}$ | $86.4^{\uparrow0.3}$ | $11.1^{\uparrow10.3}$ | $62.0^{\uparrow3.8}$ | $56.6^{\uparrow4.5}$ |
| w/o Variance-based Data Selection | 1 | 10 | 88.9 | 26.8 | 57.3 | 87.1 | 10.0 | 59.6 | 55.0 |
| **Deconta-7B-RL** | NA | NA | 94.2 | 35.7 | 66.1 | 93.4 | 1.1 | 66.7 | 59.5 |
| **+ 1-shot EM** | 1 | 10 | $95.1^{\uparrow0.9}$ | $38.0^{\uparrow2.3}$ | $68.9^{\uparrow2.7}$ | $94.1^{\uparrow0.7}$ | $10.4^{\uparrow9.3}$ | $67.0^{\uparrow0.3}$ | $62.3^{\uparrow10.2}$ |
| w/o Variance-based Data Selection | 1 | 10 | 94.4 | 36.9 | 68.0 | 93.0 | 10.8 | 67.3 | 61.7 |
| **Open Source Models** | | | | | | | | | |
| **Qwen3-8B-Base** | NA | NA | 53.8 | 22.1 | 20.4 | 31.6 | 1.2 | 49.1 | 29.7 |
| **+ 1-shot EM** | 1 | 10 | $60.4^{\uparrow6.6}$ | $26.8^{\uparrow4.7}$ | $21.3^{\uparrow0.9}$ | $35.6^{\uparrow4.0}$ | $9.2^{\uparrow8.0}$ | $54.2^{\uparrow5.1}$ | $34.6^{\uparrow4.9}$ |
| w/o Variance-based Data Selection | 1 | 10 | 60.5 | 24.3 | 21.0 | 35.2 | 8.4 | 53.8 | 33.9 |
| **Qwen2.5-Math-7B** | NA | NA | 53.0 | 11.0 | 17.2 | 44.1 | 1.0 | 48.9 | 29.2 |
| **+ 1-shot EM** | 1 | 10 | $78.8^{\uparrow25.8}$ | $35.3^{\uparrow24.3}$ | $39.7^{\uparrow22.5}$ | $70.3^{\uparrow26.2}$ | $17.4^{\uparrow16.4}$ | $65.1^{\uparrow16.2}$ | $51.1^{\uparrow24.9}$ |
| w/o Variance-based Data Selection | 1 | 10 | 77.9 | 35.0 | 37.5 | 70.5 | 11.5 | 64.3 | 49.5 |
| **Llama-3.1-Nemotron-Nano-8B-v1** | NA | NA | 69.8 | 25.7 | 37.8 | 54.7 | 0.6 | 55.2 | 40.6 |
| **+ 1-shot EM** | 1 | 10 | $75.0^{\uparrow9.2}$ | $30.1^{\uparrow4.4}$ | $38.0^{\uparrow0.2}$ | $59.3^{\uparrow4.6}$ | $8.4^{\uparrow7.8}$ | $57.9^{\uparrow2.7}$ | $44.8^{\uparrow4.2}$ |
| w/o Variance-based Data Selection | 1 | 10 | 71.3 | 29.5 | 38.3 | 60.0 | 7.9 | 57.5 | 44.1 |

softmax$(z/\tau)$, $\|\nabla_z H(p)\|_2 = \tau^{-1}\sqrt{\mathrm{Var}_{v \sim p}[\log p(v)]}$, which increases with distributional spread.

## 3. Experiment

### 3.1. Experimental Setting

We implemented the overall training process of EM based on Acclerate (Gugger et al., 2022). We selected 1 piece of data from the dataset as prompt. Since it is an unsupervised method, we do not need any data labels. We directly train the model for only 10 steps with a constant learning rate of $2 \times 10^{-5}$, a temperature of 0.5, and a batch size of 64. The reason why only 10 steps are sufficient will be explained in detail in Section 3.6.

### 3.2. Deconta 7B

To avoid unreliable conclusions arising from test set contamination in the training data (Wu et al., 2025; Shao et al., 2025), we pretrain **Deconta 7B**, a modern large language model with 7 billion parameters, entirely from scratch. Deconta 7B largely follows the architecture of OLMo 2 7B (OLMo et al., 2025), with the primary difference being the use of *pre-attention normalization* instead of the post-attention normalization adopted in OLMo 2.

For pretraining, we employ the exact same data recipe as OLMo 2 7B (OLMo et al., 2025). A detailed description of the pretraining data composition is provided in Appendix A. For post-training, we use the AceReason-Nemotron dataset (Anonymous, 2026), with full details of the post-training data pipeline described in Appendix B. Reinforcement learning fine-tuning is conducted using the Skywork-OR1-RL-Data dataset (He et al., 2025).

All training data are processed through a strict data decontamination pipeline to eliminate overlap with evaluation benchmarks. Specifically, we decontaminate against the following test sets: MATH500 (Lightman et al.,

2023), MinervaMath (Lewkowycz et al., 2022), Olympiad-Bench (He et al., 2024), AMC23, KK (Xie et al., 2024), and MBPP (Austin et al., 2021). We adopt an $n$-gram–based decontamination strategy (Brown et al., 2020). Specifically, we tokenize both training and evaluation samples into contiguous $n$-grams and compute the maximum $n$-gram overlap between each training instance and all test instances. A training sample is considered contaminated if it shares at least one identical $n$-gram with any test example. This conservative criterion is designed to eliminate both exact duplication and high-confidence paraphrastic leakage, ensuring that no training instance contains long-span lexical sequences appearing in the evaluation benchmarks.

In our experiments, we use a 10-gram threshold and remove all training samples that exhibit overlap beyond this threshold with any test instance, ensuring a rigorously decontaminated training corpus.

### 3.3. Main Result

Table 1 summarizes the main results of one-shot entropy minimization (EM) across different base models. Overall, EM demonstrates strong competitiveness against RL-based methods despite using only a single unlabeled example and minimal optimization (10 gradient steps).

On the Qwen2.5-Math-7B base model, one-shot EM yields substantial gains across all evaluated math reasoning benchmarks, improving performance by +25.8 on MATH500, +24.3 on Minerva Math, +22.5 on OlympiadBench, and +26.2 on AMC23, with an average gain of +24.9 points. Notably, these improvements significantly narrow the gap to strong RL-based baselines such as Prime-Zero-7B and RLVR-based methods, achieving a competitive score of 70.3 on AMC23.

Across different open-source model families, we observe that the effectiveness of one-shot EM is bounded by the intrinsic reasoning capacity of the base model. On Llama-3.1-Nemotron-Nano-8B-v1, EM produces consistent but more moderate improvements, with gains of +9.2 on MATH500, +4.4 on Minerva, and +4.6 on AMC23, resulting in a +4.2 increase in the overall average score. In contrast, stronger bases such as Qwen2.5-Math-7B and Qwen3-8B-Base benefit more substantially from EM, suggesting that entropy minimization primarily amplifies existing reasoning capabilities rather than creating them from scratch.

We further evaluate one-shot EM on two fully decontaminated models, Deconta-7B-SFT and Deconta-7B-RL, which provide a cleaner setting for assessing test-time adaptation effects. On Deconta-7B-SFT, EM consistently improves performance across all benchmarks, with particularly notable gains on Minerva Math (+6.3) and KK (+10.3), yielding a +4.5 improvement in the average score. On the stronger

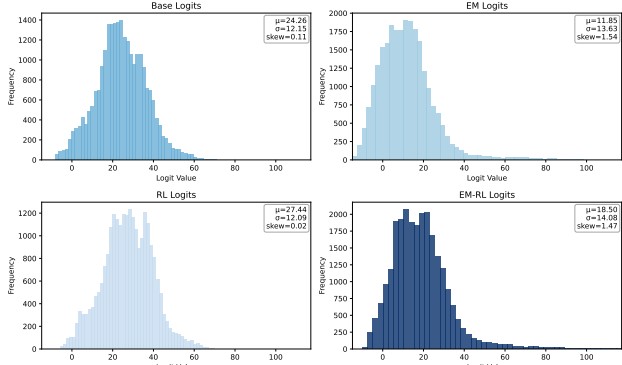

*Figure 1.* Flattened logit distributions for Base, EM, RL, and EM→RL models. EM induces a pronounced right-skew, while RL compresses the distribution leftward.

Deconta-7B-RL model, which already exhibits high baseline performance, EM continues to deliver non-trivial gains, especially on KK (+9.3), leading to a substantial +10.2 increase in the overall average score. These results indicate that one-shot EM remains effective even when applied on top of strong RL-trained models, and can be viewed as a complementary test-time optimization technique rather than a replacement for RL-based training.

We further evaluate one-shot EM on two models that are fully trained in-house and strictly decontaminated, Deconta-7B-SFT and Deconta-7B-RL, which provide a cleaner setting for assessing test-time adaptation effects. On Deconta-7B-SFT, EM consistently improves performance across all benchmarks, with particularly notable gains on Minerva Math (+6.3) and KK (+10.3), yielding a +4.5 improvement in the average score. On the stronger Deconta-7B-RL model, which already exhibits high baseline performance, EM continues to deliver non-trivial gains, especially on KK (+9.3), leading to a substantial +10.2 increase in the overall average score. These results indicate that one-shot EM remains effective even when applied on top of strong RL-trained models, and can be viewed as a complementary test-time optimization technique rather than a replacement for RL-based training.

### 3.4. Logits Shift

**Empirical Observation.** We sample 20 prompts from the NuminaMath (LI et al., 2024) dataset and generate 20 responses per model using four variants: Qwen2.5-Math-7B (Base), Qwen2.5-Math-7B-EM, Qwen2.5-Math-7B-RL, and Qwen2.5-Math-7B-EM-RL, resulting in 80 generated outputs in total. For each output, we collect all pre-softmax logits produced at every generation step.

To analyze global distributional effects, we flatten all collected logits across tokens and samples into a single multiset $\mathbf{z}_{\text{flat}}$. We quantify distributional asymmetry using the stan-

dardized skewness

$$\gamma_1 = \frac{1}{n} \sum_{i=1}^{n} \left( \frac{z_i - \mu}{\sigma} \right)^3,$$

where $\mu$ and $\sigma$ denote the mean and standard deviation of $\mathbf{z}_{\text{flat}}$. Positive skewness corresponds to a right-tailed logit distribution.

As shown in Figure 1, models trained with entropy minimization (EM) exhibit substantially increased logit skewness, indicating a pronounced right-tailed distribution. In contrast, reinforcement learning (RL) leads to a marked reduction in skewness, corresponding to a leftward compression of the logit distribution. Applying RL after EM partially reverses the right-skewness induced by EM.

**Phenomenon: Logits Shift.** We refer to this systematic redistribution of the global logit geometry as *logits shift*. A rightward logits shift corresponds to an expansion of the high-logit tail, whereas a leftward shift indicates global compression of logit mass.

**Interpretation and Contrast with RL.** A right-skewed logit distribution enlarges the set of competitive tokens during sampling, effectively expanding high-probability paths in the generation space. This reshaping promotes exploration among multiple plausible continuations.

By contrast, RL re-weights logits based on external reward signals, suppressing tokens that receive high model probability but low reward alignment. This induces global logit compression, reducing distributional skewness and contracting the sampling space. These opposing geometric effects explain the distinct logits shifts induced by EM and RL.

We now provide a theoretical explanation for why entropy minimization necessarily induces a right-skewed logit distribution.

### 3.5. Why Does Entropy Minimization Induce Right-Skewed Logits?

**Setup.** At a generation position $t$, the model produces a logit vector $z \in \mathbb{R}^{|V|}$, inducing a categorical distribution

$$p_i = \text{softmax}\left(\frac{z_i}{\tau}\right) = \frac{e^{z_i/\tau}}{\sum_j e^{z_j/\tau}},$$

where $\tau$ denotes the training temperature. The token-level entropy is

$$H(p) = -\sum_i p_i \log p_i.$$

The entropy minimization objective is

$$\mathcal{L}_{\text{EM}} = \frac{1}{|I|} \sum_{t \in I} H\left(p^{(t)}\right),$$

where $I$ indexes generated token positions.

**Proposition 1 (Gradient of Entropy w.r.t. Logits).** Let $p = \text{softmax}(z/\tau)$. Then,

$$\frac{\partial H}{\partial z_i} = \frac{1}{\tau} p_i \left( -\log p_i - H \right).$$

**Proof (Sketch).** The softmax Jacobian satisfies $\partial p_j / \partial z_i = \tau^{-1} p_j (\delta_{ij} - p_i)$, and $\partial H / \partial p_j = -(\log p_j + 1)$. Applying the chain rule and using $\sum_j p_j = 1$ and $\sum_j p_j \log p_j = -H$ yields the result. $\square$

**Corollary 1 (Thresholded Logit Update under EM).** Gradient descent on $\mathcal{L}_{\text{EM}}$ yields

$$\Delta z_i = \frac{\eta}{\tau} p_i \left( \log p_i + H \right).$$

Consequently, there exists an entropy-dependent threshold

$$p_i \gtrless e^{-H} \quad \implies \quad \Delta z_i \gtrless 0.$$

High-probability tokens receive positive logit updates, while the long tail of low-probability tokens is pushed downward. Moreover, the update magnitude is scaled by $p_i$, amplifying high-probability tokens more aggressively.

**Proposition 2 (Amplification of Logit Gaps).** For two tokens $a$ and $b$ with logit gap $\Delta_{ab} = z_a - z_b$, the continuous-time gradient flow $\dot{z}_i = -\partial H / \partial z_i$ satisfies

$$\dot{\Delta}_{ab} = \frac{1}{\tau} \left[ p_a (\log p_a + H) - p_b (\log p_b + H) \right].$$

Thus, logit gaps between high- and low-probability tokens are amplified over training.

**Lemma 1 (Increase in Logit Skewness).** Let $z \in \mathbb{R}^M$ denote a multiset of logits with skewness $\gamma_1 = \mu_3 / \sigma^3$, where $\mu_3$ is the third central moment. An EM-induced update $\Delta z_i \propto p_i (\log p_i + H)$ satisfies a thresholded tail-stretching condition: a small subset of high-probability logits is pushed upward, while the remaining majority is compressed downward. To first order,

$$\gamma_1(z + \Delta z) > \gamma_1(z).$$

**Interpretation.** Lemma 1 formalizes how entropy minimization reshapes the logit geometry: EM selectively stretches the right tail of the distribution, thereby increasing the standardized third central moment (skewness). This provides a theoretical explanation for the rightward logits shift observed empirically in Figure 1.

### 3.6. Training Loss vs. Reasoning Performance

As shown in Figure 2, the loss drops to a relatively low level around step 10, and the model's performance on mathematical reasoning reaches its peak. However, unexpectedly, as

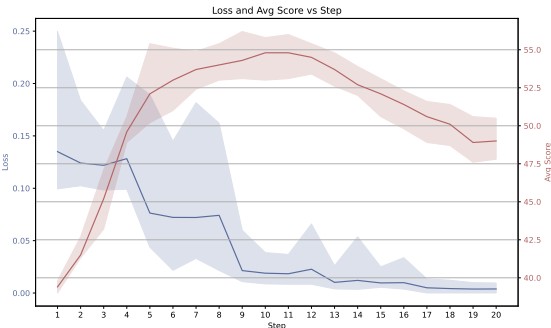

*Figure 2.* The left y-axis represents the EM training loss, while the right y-axis shows the average score across four reasoning benchmarks. It can be observed that the training loss converges rapidly, whereas the average score peaks around step 10 and then begins to decline. The results in the figure are obtained by repeating the experiments with the same training hyper-parameters using 16 different seeds to reduce randomness.

the EM training loss continues to decrease beyond step 10, the mathematical reasoning performance begins to decline. We designate this phenomenon as *over confidence*. Persistent EM may excessively amplify the model's confidence in its tokens during inference, thereby exacerbating algorithmic bias and leading to significant deviations in outputs. In conjunction with the findings presented in Section 3.4, we argue that EM functions primarily as a tool for shaping the model's distribution rather than as a learning method or strategy. Consequently, the effect of distribution shaping is largely achieved within a very small number of training steps, leading to a decoupling between the continued decrease in EM training loss and improvements in mathematical reasoning performance.

### 3.7. Sampling Temperature in Training

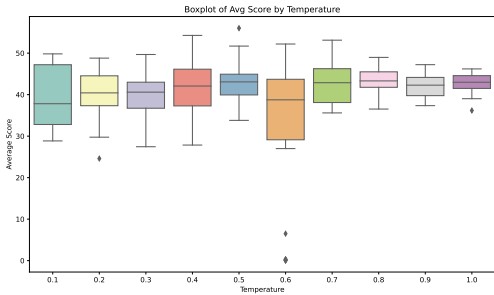

*Figure 3.* The impact of generation temperature during EM training on the average performance of the trained model across four reasoning datasets. The results in the figure are obtained by repeating the experiments with the same training hyper-parameters using 16 different seeds to reduce randomness.

As shown in Figure 3, the average performance of the EM-trained model across four math reasoning benchmarks generally exhibits an upward trend as the generation temperature

increases. The maximum of the average performance initially increases and then declines around a temperature of 0.5. Higher temperatures lead to better average reasoning ability, while moderate temperatures (e.g., 0.5) result in greater performance variance, thereby creating opportunities for higher peak performance. Therefore, we prioritize the model trained at temperature 0.5 when reporting final results.

However, as shown in the figure, EM training exhibits significant randomness. The results in the figure are obtained by repeating experiments with 16 different random seeds under the same set of hyperparameters. It can be seen that, even with identical settings, the average scores across the four math reasoning benchmarks can differ by as much as a factor of two depending on the seed. Therefore, all the conclusions in this paper are based on at least 16 repeated experiments with different seeds. We also advocate that future research should focus on reducing the stochasticity of EM training.

### 3.8. Sampling Temperature in Evaluate Generation

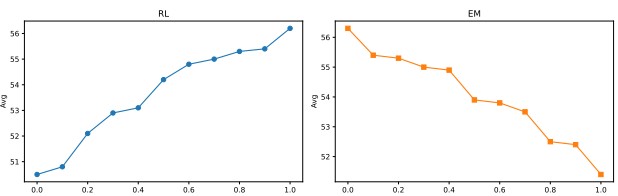

*Figure 4.* The impact of generation temperature during evalutating on the average performance of the trained model across four reasoning datasets. The results in the figure are obtained by repeating the experiments with the same training hyper-parameters using 16 different seeds to reduce randomness.

As shown in Figure 4, we observe a striking pattern in the EM-trained model's response to varying sampling temperatures during generation. Specifically, as the sampling temperature increases during evaluation, the model's average performance across four math reasoning benchmarks consistently *decreases*. This trend is in sharp contrast to that of Reinforcement Learning (RL)-trained models shown in Figure 4, where higher sampling temperatures often improve performance.

**Greedy Decoding.** The observation can be formally contextualized through the greedy decoding process, which selects the token with maximum conditional probability at each step:

$$y_t = \arg\max_{v \in \mathcal{V}} \ p_\theta(v \mid y_{<t}, x),$$

where $\mathcal{V}$ is the vocabulary and $x$ is the input prompt.

Together with our analysis in Section 3.4, we hypothesize that EM training systematically reshapes the model's logits distributions to become increasingly right-skewed. This process reinforces confidence in already high-probability tokens, effectively concentrating the probability mass on semantically coherent and correct options. As a result, greedy decoding—which deterministically selects the most probable token—becomes a highly effective strategy after EM training.

In contrast, RL adjust token probabilities based on external ground-truth reward. This often promotes the relative ranking of previously low-probability (tail) tokens. Even after reranking, these tokens tend to occupy intermediate positions in the probability distribution, requiring higher temperatures during sampling to be selected. Consequently, RL-trained models exhibit the opposite trend: performance improves with higher sampling temperatures, as seen in Figure 4.

### 3.9. EM Before/After RL

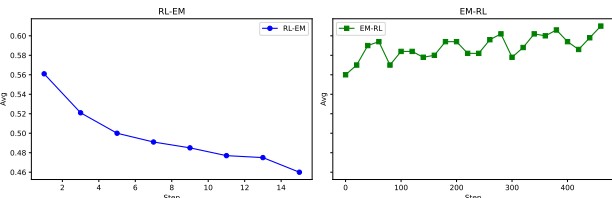

*Figure 5.* The blue curve on the left represents the average performance across four mathematical reasoning benchmarks of the model trained with RL during the EM phase as training steps progress, while the curve on the right shows the performance of the model trained with EM during the RL phase. The results in the figure are obtained by repeating the experiments with the same training hyper-parameters using 16 different seeds to reduce randomness.

Figure 5 shows a clear asymmetry: applying Entropy Minimization (EM) after Reinforcement Learning (RL) leads to a steady decline in performance across four math benchmarks, whereas applying RL after EM yields consistent gains. This suggests that EM exacerbates the distributional distortions introduced by RL, reinforcing the "alignment tax" of RL.

This aligns with prior work showing that RL is most effective when preceded by Supervised Fine-Tuning (SFT) (Casper et al., 2023; Gou & Nguyen, 2025), while applying SFT or entropy-based methods after RL often harms performance. In our case, EM after RL locks in narrow, overconfident output modes, while EM before RL enhances reasoning and allows RL to refine outputs without degrading diversity or accuracy.

## 4. Related Work

### 4.1. Entropy Minimization

Wang et al. (Wang et al., 2025) discovered that RLVR can achieve performance comparable to using thousands of data with just a single data, which is one of the main inspirations for our work. They also were the first to observe that simply optimizing the entropy loss can significantly improve reasoning performance. However, they did not explore this phenomenon in depth, stating in the original paper, "We leave the rigorous analysis to future works." Agarwal et al. (Agarwal et al., 2025) is the first to study entropy minimization during post-training for large language models. However, they considered the effectiveness of entropy minimization to be unreasonable; although they presented experimental results, their analysis of entropy minimization remained limited.

### 4.2. Reinforcement Learning for LLM

Recent research has increasingly explored post-training approaches to improve the reasoning abilities of large language models. These approaches often involve additional fine-tuning or reinforcement learning using curated datasets that include reasoning tasks and chain-of-thought annotations (Xu et al., 2025; Xie et al., 2025; Gao et al., 2024; Zeng et al., 2025; Hu et al., 2025; Cui et al., 2025; Wang et al., 2025; Agarwal et al., 2025). Reinforcement learning techniques such as Direct Preference Optimization (DPO) (Rafailov et al., 2024), Proximal Policy Optimization (PPO) (Schulman et al., 2017), Group Relative Policy Optimization (GRPO) (Shao et al., 2024), and REINFORCE++ (Hu, 2025) are gaining prominence in this area.

## 5. Conclusion

We show that one-shot entropy minimization provides a simple, fully unsupervised, and highly efficient alternative to reinforcement learning for post-training large language models. Using only a single unlabeled example and a few gradient steps, it substantially improves reasoning performance by reshaping the model's output distribution and amplifying confident reasoning trajectories.

Beyond empirical gains, we present a rigorous theoretical analysis of entropy minimization. We characterize its optimization dynamics, establish its connection to policy-gradient–style updates, and explain why high-variance prompts yield stronger training signals and induce a distinctive right-skewed logit geometry. Together, these results show that one-shot entropy minimization enhances reasoning primarily by calibrating confidence over existing knowledge, positioning it as both a practical post-training method and a principled framework for understanding latent reasoning in pretrained models.

## 6. Impact Statements

This paper presents work whose goal is to advance the field of machine learning. There are many potential societal consequences of our work, none of which we feel must be specifically highlighted here.

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

## A. Deconta-7B Pretraining Receipt

*Table 2.* Data composition of the OLMo-mix pretraining corpus.

| Name | Tokens | Bytes (Uncompressed) | Documents |
|---|---|---|---|
| DCLM-Baseline | 3.70T | 21.3TB | 2.95B |
| Arxiv | 20.8B | 77.2GB | 3.95M |
| pes2o | 58.6B | 412GB | 38.0M |
| StarCoder | 83.0B | 458GB | 78.7M |
| Algebraic-Stack | 11.8B | 44.0GB | 2.83M |
| OpenWebMath | 12.2B | 47.2GB | 2.89M |
| Wikipedia | 3.66B | 18.1GB | 6.17M |
| **Total** | **3.90T** | **22.4TB** | **3.08B** |

## B. Deconta-7B SFT Receipt

*Table 3.* Composition of the AceReason-Nemotron supervised fine-tuning (SFT) dataset.

| Source | # Questions | # Samples |
|---|---|---|
| OpenMathReasoning | 270,534 | 2,147,570 |
| NuminaMath-CoT | 78,880 | 521,171 |
| OpenCodeReasoning | 35,374 | 763,495 |
| MagicoderEvolInstruct | 27,625 | 27,625 |
| opc-sft-stage2 | 79,938 | 323,163 |
| LeetCode | 5,571 | 126,878 |
| TACO | 16,726 | 56,694 |
| APPS | 159 | 3,736 |
| **Total** | **515,807** | **3,970,332** |

