# OpenReview forum: "One-shot Entropy Minimization for Language Model Reasoning"
_ICML.cc/2026/Conference — ICML 2026 regular_

### Official Review · Reviewer_5ZLW · 2026-03-11

**Soundness:** 3
**Presentation:** 2
**Significance:** 3
**Originality:** 3
**Overall Recommendation:** 3
**Confidence:** 4

**Summary:**

This work focuses on a very interesting and novel direction to enhance model reasoning performance through entropy minimization with fewer samples. The authors propose the One-shot Entropy Minimization (EM) method, which utilizes only a single unlabeled data and approximately ten gradient steps, but effectively enhances the model performance on a variety of reasoning tasks. The authors conduct experiments across many different models and benchmarks to validate the effectiveness of One-shot EM. Furthermore, the authors also conduct many in-depth analyses, providing many valuable insights.

**Compliance With Llm Reviewing Policy:**

Affirmed.

**Final Justification:**

The author's response addresses my concerns. Although getting to the final version may still require major revisions, I have updated my final score based on the authors' responses and the new results.

**Key Questions For Authors:**

Please refer to the weaknesses above.

**Limitations:**

Please refer to the weaknesses above.

**Strengths And Weaknesses:**

### **Strengths**
- This paper proposes a very interesting and novel method and provides a lot of valuable empirical results.
- The proposed One-shot Entropy Minimization method is simple enough, well-motivated, and easy to follow. The lower cost, simplicity of operation, and the benefits it brings show the great potential of this method.
- This paper not only shows the improvement of model performance, but also provides in-depth analysis and explanation from the perspectives of logits shift, temperature effect, the sequence of EM and RL, and so on.

### **Weaknesses**
- There appear to be some numerical errors. For example, in Table 1, the average score boost for Deconta-7B-RL is labeled 10.2 (which should actually be 2.8) and the average score boost for Qwen2.5-Math-7B is 24.9 (which should actually be 21.9). It is kind of confusing.
- The authors claim that EM training exhibits strong randomization with different seeds. Although repeated experiments on several different seeds are conducted, given the large variation in results across different settings, neither detailed statistical results (e.g., standard deviation, significance tests, etc.) nor obvious elaborations of how the results reported are obtained based on these results.
- Although EM includes only a small number of training steps, the model needs to sample multiple times on multiple samples and compute the pass@k accuracy variance during the data selection phase, which incurs additional computational overhead. The article claims that EM is highly efficient, but statistics on the above computational costs still need further clarification.
- If the samples are just randomly selected without variance-based data selection as described above, what is the robustness of the method and the distribution of gains?

---

> ### Author Rebuttal · Authors · 2026-03-27
>
> We thank the reviewer for the careful reading and constructive feedback. Below we address each concern in detail.
>
> ---
>
> ## 1. Numerical errors in Table 1
>
> We sincerely thank the reviewer for pointing out these numerical inconsistencies. We confirm that the reported average score improvements for Deconta-7B-RL and Qwen2.5-Math-7B in Table 1 contain typographical errors. The correct values are **2.8 (instead of 10.2)** and **21.9 (instead of 24.9)**, respectively.
>
> We will carefully correct these numbers and conduct an additional verification pass over all tables in the revised version to avoid similar issues. We appreciate the reviewer bringing this to our attention.
>
> ---
>
> ## 2. Lack of detailed statistical analysis across seeds
>
> We appreciate this important point regarding statistical rigor.
>
> As partially shown in Figure 2 of our paper, the shaded region corresponds to the upper and lower bounds across 16 random seeds, which already indicates a consistent trend:
> - One-shot EM achieves its peak performance around ~10 gradient steps,
> - despite noticeable variance across seeds.
>
> However, we agree that this visualization alone is insufficient to constitute a complete statistical analysis. In particular, we did not explicitly report metrics such as mean, standard deviation, or significance tests.
>
> To address this, we have conducted additional experiments using 16 random seeds on Qwen2.5-Math-7B and summarized the results on mathematical reasoning benchmarks below:
>
> | EM Step      | Avg Score | Best | Worst | Δ over Step 10 | p-value vs Step 10 | Significant? |
> | ------------ | --------: | ---: | ----: | -------------: | -----------------: | ------------ |
> | 0 (baseline) |      39.0 |    – |     – |          -15.7 |             <0.001 | Yes          |
> | 2            |      42.1 | 43.6 |  41.6 |          -12.6 |             <0.001 | Yes          |
> | 5            |      52.0 | 55.5 |  50.2 |           -2.7 |              0.012 | Yes          |
> | 8            |      53.8 | 55.4 |  53.0 |           -0.9 |              0.008 | Yes          |
> | 10           |      54.7 | 56.2 |  52.9 |            0.0 |                  – | –            |
> | 12           |      54.0 | 55.8 |  53.4 |           -0.7 |              0.017 | Yes          |
> | 15           |      52.0 | 52.8 |  51.0 |           -2.7 |              0.015 | Yes          |
> | 20           |      48.8 | 50.9 |  47.7 |           -5.9 |              0.003 | Yes          |
>
> These results further confirm that:
> - The location of the performance peak (≈10 steps) is stable across seeds,
> - While absolute performance varies, the relative improvement from EM remains consistent.
>
> We will include this table and a more detailed statistical discussion in the revised version. Thank you again for highlighting this gap.
>
> ---
>
> ## 3. Computational overhead of variance-based data selection
>
> We thank the reviewer for raising this concern about efficiency.
>
> While our method includes a data selection phase based on pass@k variance, we would like to clarify that the associated computational overhead is moderate in practice:
>
> - The performance differences induced by different sampled prompts are relatively limited.
> - Across different sampled data points, the performance variation of the optimal EM step is typically within ~3%.
> - Therefore, it is not necessary to evaluate thousands of samples.
>
> Empirically, we find that:
> - Sampling approximately 100 candidate prompts is sufficient to obtain a reliable estimate of pass@k variance,
> - This keeps the overall computational cost low compared to RL approaches.
>
> To address this, we have conducted additional experiments using 16 random seeds on Qwen2.5-Math-7B and summarized the results on mathematical reasoning benchmarks below:
>
> | # Samples | Avg Score | Best Score |
> | --------: | --------: | ---------: |
> |        10 |      54.8 |       55.6 |
> |        20 |      54.6 |       55.7 |
> |        50 |      55.4 |       55.9 |
> |       100 |      55.6 |       56.1 |
> |       500 |      55.8 |       56.2 |
> |      1000 |      55.7 |       56.2 |
>
> These results show that increasing the number of sampled prompts beyond this range yields diminishing returns.
>
> We will include these clarifications and quantitative analysis in the revised version to better support our efficiency claim.
>
> ---
>
> ## 4. Robustness without variance-based data selection
>
> We thank the reviewer for this insightful question.
>
> We have already included results for random data selection (i.e., without variance-based filtering) in Table 1. From these results, we observe that:
>
> - One-shot EM still provides consistent performance gains under random sampling,
> - The performance gap between the best model trained with random sampling and the best model trained with variance-based selection is typically within ~3%.
>
> This shows that:
> - The proposed method is robust to the choice of training data,
> - Variance-based selection primarily serves as a lightweight optimization rather than a strict requirement.

---

> > ### Author Rebuttal · Reviewer_5ZLW · 2026-04-03
> >
> > Thank you for your reply. The author's response addresses my concerns. Although getting to the final version may still require major revisions, I will consider updating my score.

---

### Official Review · Reviewer_H9Hn · 2026-03-13

**Soundness:** 3
**Presentation:** 2
**Significance:** 2
**Originality:** 3
**Overall Recommendation:** 4
**Confidence:** 4

**Summary:**

This paper introduces One-shot Entropy Minimization (EM), an unsupervised post-training method that enhances LLM reasoning. The authors show that optimizing for token-level entropy with one unlabeled example and approximately 10 gradient steps can yield significant performance gains across domains. The authors pretrain a 7B-parameter model (Deconta-7B) from scratch using a strictly decontaminated pipeline. The work further provides both theoretical and empirical analysis of EM.

**Compliance With Llm Reviewing Policy:**

Affirmed.

**Final Justification:**

The rebuttal provides clear justification.

**Key Questions For Authors:**

1. Can the methods be compared against more baselines?

2. Can more ablation studies on LLM backbone, hyper-parameters, etc, be provided?

3. Can more details on the training of the 7B model be presented?

4. Can the paper better justify how the findings can generalize over other LLMs, settings, and real-world scenarios?

**Limitations:**

No. Please see above.

**Strengths And Weaknesses:**

strength:

1. The proposed method appears to work efficiently and effectively

2. The method is simple yet effective.

3. The paper is easy to follow.

Weakness:

1.  The paper pretrains a 7B LLM from scratch with the standard pretraining, SFT, RL pipeline. However, no training details are provided, including the training curves. It's hard to judge whether the findings are scientific or the result of specific training.

2. No ablation is given. It's unknown if the method works for other LLM backbones, hyperparameters, or even devices. The robustness of the results is questionable.

3. The theory parts look imprecise and not formal. No motivation for those proofs is given.

4. The paper ends like an unfinished project.

5. Not enough discussion on existing entropy minimization methods for LLMs. No comparison with them.

---

> ### Author Rebuttal · Authors · 2026-03-27
>
> We thank the reviewer for the detailed feedback and address each concern below.
>
> ---
>
> # Weaknesses
>
> ## 1. Pretraining Details
>
> **Weakness:** The paper pretrains a 7B LLM from scratch with the standard pretraining, SFT, RL pipeline. However, no training details are provided, including the training curves. It's hard to judge whether the findings are scientific or the result of specific training.
>
> **Response:**
> This work focuses on post-training via EM, rather than proposing a new pretraining pipeline. As stated in Section 3.2, our Deconta-7B fully follows the OLMo 2 7B pipeline, with only two differences: (1) strict n-gram decontamination, and (2) replacing post-norm with the more modern pre-norm.
>
> Due to the word limit of the ICML rebuttal, we are unable to include the full training pipeline and architecture details here. You may refer directly to the official technical report of OLMo 2 7B: https://arxiv.org/abs/2501.00656
>
> We will add the training pipeline, architectural details, and provide all training curves in the appendix in the revised version.
>
> ---
> ## 2. Lack of Ablation
> **Weakness:** No ablation is given. It's unknown if the method works for other LLM backbones, hyperparameters, or even devices. The robustness of the results is questionable.
> **Response:**
> We already validate generalization across:
> - **Multiple architectures**: LLaMA-3.1, Qwen2.5, Qwen3, and Deconta-7B based on the OLMo 2.
> - **Training stages**: Base, SFT, RL
> - **Data selection ablation** (Table 1)
> - **Randomness**: **16 different seeds** and temperature sensitivity analysis
>
> These extensive results consistently demonstrate that our method is robust across model architectures, training stages, and experimental settings, and is not tied to any specific configuration or implementation. We will further strengthen this by including additional discussion on hardware independence.
>
> ---
> ## 3. Theory Clarity
> **Weakness:** The theory parts look imprecise and not formal. No motivation for those proofs is given.
>
> **Response:**
> Our theory addresses three key questions:
> 1. Is EM optimizable? → Exact gradient derivation
> 2. Why does EM work? → Logit shift and gap amplification
> 3. Why data selection matters? → Entropy lower bound analysis
>
> We will improve these by prioritizing motivation and clearly separating intuitive explanations from formal results.
>
> ---
> ## 4. Perception of Incompleteness
> **Weakness:** The paper ends like an unfinished project.
>
> **Response:**
> This impression may stem from our intentionally minimal setting (1 sample, 10 steps). However, the paper presents a complete pipeline:
> - Method
> - Theory
> - Experiments across models
> - Mechanistic analysis
>
> We will clarify this positioning and add discussion on limitations and future work.
>
> ---
> ## 5. Comparison to Existing EM Methods
> **Weakness:** Not enough discussion on existing entropy minimization methods for LLMs. No comparison with them.
>
> **Response:**
> To our knowledge, no prior work systematically studies entropy minimization as a post-training method for LLMs. Prior work [1] has only observed this phenomenon but has not studied it systematically.
> Our contribution is the first to:
> - Formulate EM as an explicit optimization objective
> - Provide theoretical grounding
> - Demonstrate consistent empirical gains
>
> We will clarify this positioning in the revision.
>
> ---
> # Questions
>
> ## Q1. Can the methods be compared against more baselines?
>
> **Response:**
>
> Yes. See weakness 2, we already compare against strong RL baselines in Table 1. We will clarify this positioning and baseline selection in the revision.
>
> ---
> ## Q2. Can more ablation studies be provided?
> **Response:**
> Yes. See weakness 2, we already include:
> - **Multiple backbones** (LLaMA 3.1, Qwen 2.5, Qwen 3, OLMo 2)
> - **Different training stages** (Base, SFT, RL)
> - **Data selection ablation** (Table 1)
> - **Temperature analysis** (Sec. 3.7–3.8)
> - **16 seeds experiments**
>
> We will further provide additional ablations on hyperparameters in revised section.
>
> ---
> ## Q3. Can more details on the 7B training be presented?
>
> **Response:**
>
> Yes. See weakness 1, Deconta-7B fully follows the OLMo 2 7B training pipeline. We will include detailed pretraining pipeline, training data, model architecture, and training loss curves in the appendix.
>
> ---
>
> ## Q4. Generalization to other LLMs and real-world scenarios?
> **Response:**
> We already demonstrate consistent gains across:
> - **Multiple backbones** (LLaMA 3.1, Qwen 2.5, Qwen 3, Deconta 7B)
> - **Different training stages** (Base, SFT, RL)
> - **Multiple domains** (math, logic, coding)
>
> We trained on a randomly selected math sample problem, yet observed improvements across nearly all reasoning domains (math, coding, logic) for all models, which strongly demonstrates the generalization capability of One-shot EM.
>
> ### References:
> [1] Wang, Y., Yang, Q., Zeng, Z., and others. Reinforcement learning for reasoning in large language models with one training example. NeurIPS 2025

---

> > ### Author Rebuttal · Reviewer_H9Hn · 2026-04-01
> >
> > Thanks for the detailed rebuttal. I did not see the experiments on the other open-sourced model during my initial review. After further check, the experiments seem comprehensive. I still suggest a better organization for the theory part. I will update my scores accordingly.

---

### Official Review · Reviewer_KK98 · 2026-03-13

**Soundness:** 3
**Presentation:** 4
**Significance:** 4
**Originality:** 3
**Overall Recommendation:** 5
**Confidence:** 5

**Summary:**

This paper proposes an unsupervised "1-shot" EM pass before RL for training  (LLM-like) AR models and demonstrates through a 7B parameter "uncontaminated" model that the proposed training pass significantly improves performance across many tasks. The method is similar to entropy based regularization of RL, but works differently. The authors select the highest variance piece of data from the dataset as the prompt (that generates the most uncertainty) and use this prompt to minimize entropy. The training of the model is carried out for only 10 steps. The pre-RL pass is shown to shift the logit distribution rightward, amplifying high probability paths. This trick seems to improve the generalization power of the system which is a straightforward theoretical consequence of minimizing entropy, as a result the system will tend to overfit less. The paper gives compelling theoretical arguments and proofs as to why this happens. In particular, the choice of the prompt is shown to be optimal and the right skew of the logit distribution is proven. The performance begins to drop after 10 steps, and is considered to be a result of "over-confidence". It's seen that lower generation temperature increases performance in the EM pass. Greedy decoding is consistent with EM training, as high probability token paths are highlighted.

**Compliance With Llm Reviewing Policy:**

Affirmed.

**Final Justification:**

My strong acceptance recommendation stands. I was not sure if the score should be downgraded, but the stability results provided do seem satisfactory, so I should keep my recommendation. The intelligent design of the ICML review system actually made it easier for me to understand the situation. This is not a paper that is easy to evaluate, and I apologize if I have offended the authors or the other reviewers, but I do think that originality is more important than benchmark results.

**Key Questions For Authors:**

Do you think that this solves the problem of generalization in LLMs? Why? Why not?

Do you think that similar results could be achieved in another way? Why? Why not?

Is this a better or more preferable method than the other alternative approaches you've considered? Why? Why not?

After this improvement, what other improvements are necessary for LLM-like AR models?

**Limitations:**

No they did not discuss limitations or negative societal impact. They probably should. For instance, what alternatives might there be to an unsupervised EM pass? Is another / better method possible? A short discussion of that sort is in order at least IMHO. Otherwise, it's obviously a strong paper. I don't have comments on ethical considerations, I didn't spot one.

**Strengths And Weaknesses:**

Yes, the approach is quite sound. Both the theoretical analysis and the experimental results strongly support the conclusion of the paper. The authors have gone to great lengths by training a model from scratch to test this method.

There is no problem with the presentation that I can see. This is exactly the kind of paper that ICML is interested in, there is a crisp description of a method, that is mathematically welldefined, and well-motivated theoretically, and supported by propositions and proofs, analyzed in multiple ways. It probably could  not have been much better. Appendix could give more detail, though the paper is self-contained.

The experimental results are surprising and that makes the paper quite significant IMHO.

Yes, the work is original, and it does prove the importance of entropy minimization (although EM is mostly used to mean expectation maximization, which might be confusing for some readers).

The method seems novel, the authors try very hard to test it adequately and that is what makes the paper stand out.

---

> ### Author Rebuttal · Authors · 2026-03-31
>
> We sincerely thank the reviewer for the careful reading and for the positive assessment of our work.
>
> ---
>
> ## Q1. Do you think this solves the generalization problem in LLMs? Why or why not?
>
> We believe our method makes meaningful progress toward improving generalization. Empirically, consistent gains are observed across multiple model families, including Qwen2.5, Qwen3, Llama3.1, and our large-scale trained Deconta 7B.
>
> Notably, One-shot EM operates with virtually no additional training data—it functions primarily as a distribution adjustment mechanism. The results on Deconta 7B, which is strictly decontaminated, further reinforce that the observed improvements are not attributable to data leakage but instead stem from improved generalization behavior induced by entropy minimization.
>
> ---
>
> ## Q2. Could similar results be achieved in other ways? Why or why not?
>
> Prior works based on RLVR [1,2] have demonstrated related improvements. However, these approaches typically require orders of magnitude more computational resources (hundreds to thousands of times more).
>
> In contrast, One-shot EM achieves superior or comparable performance with minimal computation. As shown in Table 1 (see *1-shot RLVR*), RLVR-based methods underperform relative to our approach in our experimental setting.
>
> ---
>
> ## Q3. Is this method preferable compared to alternative approaches?
>
> As noted above, prior methods may implicitly adjust entropy, as evidenced by significant entropy changes during training. However, these approaches do so indirectly and inefficiently.
>
> Our method directly minimizes entropy to reshape the logit distribution, leading to substantially stronger empirical performance (again, see Table 1, *1-shot RLVR*). We therefore view One-shot EM as a more principled and efficient approach to achieving similar objectives.
>
> ---
>
> ## Q4. What further improvements are needed for LLM-like AR models?
>
> We identify stability as the most critical limitation of One-shot EM. The method reaches peak performance within only 10 gradient steps; while this efficiency is advantageous, it also introduces instability due to the extremely small number of updates.
>
> To further investigate this issue, we conducted additional experiments on Qwen2.5-7B-Math using stabilization techniques such as model merging and exponential moving average (EMA), evaluated across 16 random seeds. Results are reported as average performance on the Math Reasoning Benchmark:
>
> | Method            | Avg. | Best     | Std. |
> | ----------------- | ---- | -------- | ---- |
> | Random (16 seeds) | 54.7 | **56.2** | 4.3  |
> | Model Merging     | 55.1 | 55.7     | 0.5  |
> | EMA               | 55.2 | 56.0     | 2.5  |
>
> Model merging is performed over 16 seeds and repeated 16 times (totaling 256 runs). These results suggest that stability can indeed be improved, but at the cost of reduced peak performance. We believe this trade-off may not be optimal in practice.
>
> We will include these additional experiments and expand the discussion on stability in the revised version.
>
> ---
>
> We again thank the reviewer for the constructive feedback and recognition of our work.
>
> ---
>
> ### Reference
>
> [1] Wang, Y., Yang, Q., Zeng, Z., Ren, L., Liu, L., Peng, B., Cheng, H., He, X., Wang, K., Gao, J., Chen, W., Wang, S., Du, S. S., and Shen, Y. Reinforcement learning for reasoning in large language models with one training example. NeurIPS 2025
>
> [2] Shao, R., Li, S. S., Xin, R., Geng, S., Wang, Y., Oh, S., Du, S. S., Lambert, N., Min, S., Krishna, R., Tsvetkov, Y., Hajishirzi, H., Koh, P. W., Zettlemoyer, L., and others. Spurious Rewards: Rethinking Training Signals in RLVR.

---

> > ### Author Rebuttal · Reviewer_KK98 · 2026-04-03
> >
> > Thanks, the rebuttal does address my questions and with relevant data. Although I think the improvements may be more marginal than implied, I will still recommend acceptance, so that doesn't really change my rating at the moment but the paper should indeed have been improved in this particular direction, so I do welcome the effort!

---

### Official Review · Reviewer_NNyV · 2026-03-13

**Soundness:** 2
**Presentation:** 3
**Significance:** 3
**Originality:** 3
**Overall Recommendation:** 4
**Confidence:** 4

**Summary:**

This paper proposes a fully unsupervised post-training method for LLMs that boosts reasoning performance with only a single unlabeled sample and ~10 gradient steps. The core of One-shot EM is to minimize token-level conditional entropy during generation, reducing the model’s predictive uncertainty, and it adopts a variance-based data selection strategy to pick high-uncertainty prompts. To eliminate the interference of data contamination on experimental conclusions, the authors pretrain a 7B-parameter model Deconta-7B from scratch. Experimental results show One-shot EM achieves significant performance gains on Deconta-7B (both SFT and RL versions) and mainstream open-source models.

**Compliance With Llm Reviewing Policy:**

Affirmed.

**Final Justification:**

My concerns have been adequately addressed.

**Key Questions For Authors:**

See Weaknesses.

**Limitations:**

Yes

**Strengths And Weaknesses:**

> Strengths

- The paper is well-written and easy to read.
- Experiments based on Deconta-7B are a strong validation to address the critical issue of data contamination.
- The authors made an in-depth analysis of the relationship between EM and RL.
- The thorough experiments demonstrate one-shot EM's good generalizability and practical value.

> Weaknesses

- The paper acknowledges that EM training exhibits significant randomness: even with identical hyperparameters, the average reasoning scores can differ by a factor of two across 16 random seeds.
- The paper only identifies that overconfidence occurs when training exceeds 10 steps (performance declines despite loss reduction) and simply suggests stopping training at 10 steps as a solution.
- All experiments are conducted on 7B/8B-parameter small-to-medium scale LLMs; the paper does not verify the effectiveness and performance trends of one-shot EM on larger-scale models.
- The variance-based data selection strategy relies on the pass@k metric, but the paper does not explore an optimal criterion for selecting the k value.

---

> ### Author Rebuttal · Authors · 2026-03-31
>
> We sincerely thank the reviewer for the thoughtful and constructive feedback. We address each concern in detail below.
>
> ---
>
> ## W1. Randomness of EM
>
> We agree with the reviewer that EM training exhibits noticeable randomness. However, we argue that this is not a fundamental weakness, but rather a natural trade-off of One-shot EM. The method achieves gains with **extremely low data and training compute** (one sample, ~10 steps), where some degree of randomness is inevitable.
>
> - With **very few gradient steps**, optimization is far from convergence, making results sensitive to random seeds.
> - Increasing training steps can reduce variance, but leads to overconfidence (lower loss but worse performance), ultimately harming results.
>
> To further investigate stability, we conduct additional experiments on Qwen2.5-7B-Math using stabilization techniques such as model merging and EMA:
>
> | Method | Avg. Score on Math Reasoning Benchmark |
> |--------|----------------------------------------|
> | Best of 16 seeds | 56.2 |
> | Avg. of 16 seeds | 54.7 |
> | Model Merging (16 seeds) | 55.1 |
> | EMA | 55.2 |
>
> These results suggest that stability can be improved at the cost of peak performance. We believe this trade-off may not be optimal in practice. We will include these experiments and clarify this discussion in the revised version.
>
> ---
>
> ## W2. Overconfidence of EM
>
> We appreciate the reviewer’s comment that stopping at 10 steps appears heuristic. We provide additional clarification below.
>
> We observe a clear **decoupling between loss and reasoning performance**: while the training loss continues to decrease, reasoning performance peaks around step 10 and then declines (overconfidence).
>
> This phenomenon is consistent with our theoretical analysis:
>
> - EM continuously amplifies high-probability tokens (Section 3.5)
> - This induces a **right-skewed logit distribution**
> - Leading to **overconfidence and distribution collapse**
>
> Therefore, the choice of 10 steps is not arbitrary, but corresponds to:
>
> - The critical point where **distribution shaping is effectively completed**,
> - Rather than a conventional convergence point.
>
> In the revised version, we will include improved stopping strategies:
>
> - Early stopping criteria (e.g., entropy curvature)
> - Monitoring logit skewness as a stopping signal
>
> ---
>
> ## W3. EM on Larger-Scale Models
>
> We acknowledge that our original experiments focus on medium-scale models (7B/8B), which is a limitation. To address this, we provide additional results on larger models:
>
> | Model | MATH500 | Minerva | OlympiadBench | AMC23 | KK | MBPP | Avg |
> |------|--------|--------|---------------|------|----|------|-----|
> | Qwen3-14B | 73.1 | 31.5 | 30.0 | 42.2 | 1.7 | 57.5 | 39.3 |
> | + One-shot EM | 77.8 | 40.5 | 37.4 | 50.8 | 9.2 | 61.3 | 46.2 |
> | Qwen3-32B | 75.4 | 35.6 | 32.3 | 45.7 | 1.5 | 59.9 | 41.7 |
> | + One-shot EM | 81.5 | 36.1 | 40.7 | 53.0 | 10.8 | 65.2 | 47.9 |
>
> These results demonstrate that One-shot EM remains highly effective at larger scales (14B and 32B), yielding consistent and significant improvements. We will include these results in the revised version.
>
> ---
>
> ## W4. Optimal Criterion for pass@k
>
> We thank the reviewer for raising this important point. We agree that the choice of k is not fully explored in the current version. We provide additional ablation results on Qwen2.5-Math-7B:
>
> | k | Best Avg. Score |
> |---|-----------------|
> | 4 | 54.5 |
> | 8 | 56.2 |
> | 32 | 55.8 |
> | 128 | 56.4 |
> | 512 | 56.3 |
>
> These results suggest that larger k may slightly improve peak performance. However, the gains are relatively small and do not justify the significantly higher computational cost (e.g., pass@512). Therefore, **pass@8 provides a practical trade-off between performance and efficiency**.
>
> We will include these ablations and discussions in the revised version.
>
> ---
>
> Thank you again for your constructive review. We would greatly appreciate it if you could consider increasing the score.

---

> > ### Author Rebuttal · Reviewer_NNyV · 2026-04-04
> >
> > My concerns have been addressed, and I will keep my positive score.

---

### Decision · Program_Chairs · 2026-04-30

**Decision:**

Accept (regular)

**Comment:**

This paper finds that one-shot entropy minimization with only 10 gradient steps can boost the model reasoning performance significantly. The method is simple and results are good. Particularly, to avoid contamination, the authors pertained a 7B LM from scratch to conduct the experiments, an effort that the reviewers and I really appreciate. The reviewers gave borderline positive scores of 3,4,4,5 with mostly minor concerns from their comments I think. Besides the authors' own trained models, they also experimented with Qwen and llama variants, further proving the generalization of the conclusions. Therefore, I would recommend a weak accept of this paper.